# Photonic Label-Free Biosensors for Fast and Multiplex Detection of Swine Viral Diseases

**DOI:** 10.3390/s22030708

**Published:** 2022-01-18

**Authors:** Maribel Gómez-Gómez, Carles Sánchez, Sergio Peransi, David Zurita, Laurent Bellieres, Sara Recuero, Manuel Rodrigo, Santiago Simón, Alessandra Camarca, Alessandro Capo, Maria Staiano, Antonio Varriale, Sabato D’Auria, Georgios Manessis, Athnasios I. Gelasakis, Ioannis Bossis, Gyula Balka, Lilla Dénes, Maciej Frant, Lapo Nannucci, Matteo Bonasso, Alessandro Giusti, Amadeu Griol

**Affiliations:** 1Nanophotonics Technology Center, Universitat Politècnica de València, 46022 València, Spain; dazuher@ntc.upv.es (D.Z.); blaurent@ntc.upv.es (L.B.); agriol@upvnet.upv.es (A.G.); 2Lumensia Sensors S.L., 46022 València, Spain; csanchez@lumensia.com (C.S.); speransi@lumensia.com (S.P.); srecuero@lumensia.com (S.R.); mrodrigo@lumensia.com (M.R.); ssimon@lumensia.com (S.S.); 3Institute of Food Science, National Research Conuncil, 83100 Avelino, Italy; alessandra.camarca@isa.cnr.it (A.C.); alessandro.capo@isa.cnr.it (A.C.); maria.staiano@isa.cnr.it (M.S.); antonio.varriale@isa.cnr.it (A.V.); sabato.dauria@cnr.it (S.D.); 4URT-ISA at Department of Biology, University of Naples Federico II, 80126 Napoli, Italy; 5Department of Biology, Agriculture and Food Sciences, National Research Council of Italy (CNR-DISBA), 00185 Rome, Italy; 6Laboratory of Anatomy and Physiology of Farm Animals, Department of Animal Science, Agricultural University of Athens (AUA), 11855 Athens, Greece; gmanesis@aua.gr (G.M.); gelasakis@aua.gr (A.I.G.); 7Laboratory of Animal Husbandry, Department of Animal Production, School of Agriculture, Faculty of Agriculture, Forestry and Natural Environment, Aristotle University of Thessaloniki, 54124 Thessaloniki, Greece; bossisi@agro.auth.gr; 8Department of Pathology, University of Veterinary Medicine, István u. 2, 1078 Budapest, Hungary; balka.gyula@univet.hu (G.B.); denes.lilla@univet.hu (L.D.); 9Department of Swine Diseases, National Veterinary Research Institute, al. Partyzantow 57, 24-100 Pulawy, Poland; maciej.frant@piwet.pulawy.pl; 10Dipartimento di Scienze e Tecnologie Agrarie Alimentari Ambientali e Forestali (DAGRI), Università degli Studi di Firenze, 50144 Florence, Italy; lapo.nannucci@gmail.com; 11Kontor 46 SaS, 10123 Turin, Italy; matteo.bonasso@kontor46.eu; 12CyRIC, Cyprus Research and Innovation Centre, 2414 Nicosia, Cyprus; info@cyric.eu

**Keywords:** swine disease, photonics, antibody, ring resonator, photonic integrated circuit (PIC), biosensor, label-free

## Abstract

In this paper we present the development of photonic integrated circuit (PIC) biosensors for the label-free detection of six emerging and endemic swine viruses, namely: African Swine Fever Virus (ASFV), Classical Swine Fever Virus (CSFV), Porcine Reproductive and Respiratory Syndrome Virus (PPRSV), Porcine Parvovirus (PPV), Porcine Circovirus 2 (PCV2), and Swine Influenza Virus A (SIV). The optical biosensors are based on evanescent wave technology and, in particular, on Resonant Rings (RRs) fabricated in silicon nitride. The novel biosensors were packaged in an integrated sensing cartridge that included a microfluidic channel for buffer/sample delivery and an optical fiber array for the optical operation of the PICs. Antibodies were used as molecular recognition elements (MREs) and were selected based on western blotting and ELISA experiments to ensure the high sensitivity and specificity of the novel sensors. MREs were immobilized on RR surfaces to capture viral antigens. Antibody–antigen interactions were transduced via the RRs to a measurable resonant shift. Cell culture supernatants for all of the targeted viruses were used to validate the biosensors. Resonant shift responses were dose-dependent. The results were obtained within the framework of the SWINOSTICS project, contributing to cover the need of the novel diagnostic tools to tackle swine viral diseases.

## 1. Introduction

Widespread disease outbreaks can cause serious direct economic consequences to the livestock industry of affected countries and induce multi-sectoral impacts such as trade barriers and increased food prices [1]. Establishment of control measures on time is crucial to mitigate the spread of the disease and to limit socio-economic consequences. Therefore, the need of reliable, cost-effective solutions for early detection of livestock diseases that can be translated into diagnostic devices able to be deployed directly on the field while keeping the analytical quality of commercial laboratories, has emerged [2]. The international research consortium on animal health STAR-IDAZ (https://www.star-idaz.net/priority-topic/, accessed on 10 July 2021) has also listed the development of efficient diagnostic tools among top research priorities. Based on their pathogenicity, transmission dynamics, and economical significance, the most important porcine viral diseases in Europe are classified in two major categories: (i) notifiable diseases listed by the World Organization for Animal Health (OIE), such as African Swine Fever (ASF), Classical Swine Fever (CSF) and Porcine Reproductive and Respiratory Syndrome (PRRS) [3,4,5]; and (ii) endemic diseases not listed by the OIE, but having a significant impact on swine industry, such as Porcine Circovirus type 2 (PCV2) Associated Diseases (PCVADs), Swine Influenza (SIV) and reproductive failure due to Porcine Parvovirus (PPV) infections [6,7,8].

Traditionally, the detection of these viruses relies on different approaches: direct detection of the pathogen’s genome by Polymerase Chain Reaction (PCR), its antigens by Enzyme-Linked Immunosorbent Assay (ELISA), or the whole, infectious virus by cell culture-based virus isolation, and indirect methods such as the detection of circulatory antibodies in blood samples. However, these methods require trained personnel, expensive and non-portable equipment, and a long time to provide results [9,10]. To overcome these limitations, different strategies for detecting infectious animal diseases have been explored [11]. These strategies include the development of biosensors based on optical techniques, electrochemical detection, acoustic methods, Mass Spectroscopy (MS), Nuclear Magnetic Resonance (NMR), and Magneto-Resistive Sensors (GMR), which are integrated into Lab-on-Chip (LOC) devices for the detection of pathogens [12]. In the last years, a multitude of novel sensors and devices have been developed as a response to the increased demand for analytical tools that allow detection and characterization of analytes beyond the boundaries of traditional laboratories. Among those sensors, Photonic Integrated Circuits (PICs) have the potential to detect analytes with a sensitivity and specificity previously not realized.

In this work, we report the label-free, direct detection of the six targeted viruses causing the aforementioned diseases using an optofluidic sensor, based on PICs, in particular, based on Resonant Rings (RRs) as transducers. RRs were fabricated in silicon nitride and functionalized with antibodies to capture the targeted viral antigens. The objectives of this study were (i) to assess the performance of commercial antibodies used as Molecular Recognition Elements (MREs), (ii) to biofunctionalize RRs with the selected MREs, and (iii) to detect the targeted viruses with the novel biosensors using reference samples (cell culture supernatants) as proof of concept [13].

## 2. Materials and Methods

### 2.1. Biosensors Description

The photonic biosensor developed in this research is based on the use of silicon nitride ring resonators as transducers. The notch resonances of their transmission spectra are used to detect refractive index changes near their surfaces. It is the light–matter interaction between the evanescent field of the circulating optical mode inside the ring resonator and the molecules close to the surface of the transducers that gives rise to notch resonance shifts. Wave-length-interrogation is adopted to monitor the shifts that the notch resonance undergoes when the environment near the surface of the sensor changes. The biosensor Photonic Integrated Circuit (PIC) consists of eight ring resonators (RRs) [13], as shown in Figure 1a. The whole PIC is covered by a SiO_2_ layer (1000 nm width) to optimize the performance of the Grating Couplers (GC), and, in addition, as a protective layer. A window is opened around the RRs to allow access for the functionalization of their surfaces and their exposure to the sample, as shown in Figure 1b. The fabrication details of the developed PICs are further described in an authors’ previous work [13].

### 2.2. Identification, Purification and Characterization of the MREs

Antibodies were used as molecular recognition elements (MREs) for the surface functionalization of the RRs biosensor. The selection of the MREs for the detection of the viruses targeted by the SWINOSTICS project was performed following a common workflow: (i) research on literature, (ii) purchase of commercial antibodies, (iii) purification by affinity chromatography, from serum or ascites fluid, and (iv) immunological characterization by Enzyme-Linked Immunosorbent Assay (ELISA) and Western Blot (WB) tests. The identification and selection of viral antigens were performed after studying the genetics, morphology, biochemistry, and clinical and epidemiological features of each virus and selecting the most antigenic proteins of the virions (as antigens) found in the main virus serotypes circulating in the European territory.

For each virus, the recombinant antigen virus proteins and the commercial polyclonal (pAb) and monoclonal antibodies (mAb) able to recognize the selected virus proteins were identified. The recombinant antigen proteins were used as positive controls in the immunological experiments.

#### 2.2.1. Purification

Rabbit sera or ascites fluids were diluted 1:1 with binding buffer sodium phosphate (20 mM), pH 7.0, applied to a protein A column (nProtein A Sepharose 4 Fast Flow, GE Healthcare, Chicago, IL, USA), and incubated overnight at 4 °C on a rotating wheel. Afterwards, the flow-through was removed and the column was washed with 10 column volumes of binding buffer. This step allows to remove all non-bonding proteins. Subsequently, the IgG fraction was purified following the manufacturer’s instructions. The IgG fraction was eluted with glycine (0.1 M) at pH 3.0, and immediately buffered in Tris-HCl (1.0 M), at pH 9.0, and its concentration and purity were checked by absorbance measurement at λ = 278 nm and Sodium Do-Decyl Sulfate-Polyacrylamide Gel Electrophoresis (SDS-PAGE) analysis, respectively. At the next step, the amine contained in the sample due to Tris-HCl was removed by centrifugal filtration with a Viva spin 500 10 KDa (Sartorius^TM^) and several dialysis steps by a semi-permeable membrane (2500 Dalton cut-off).

#### 2.2.2. Western Blotting Experiments

WB experiments were performed as described by Varriale et al. [14]. In brief, aliquots of 15 µL of virus samples, recombinant antigens, and monoclonal and polyclonal antibodies were heated at 95 °C in the presence of Laemmli buffer [15] and were separated on a sodium dodecyl sulfate polyacrylamide gel electrophoresis (12% SDS-PAGE). After separation, the proteins were transferred onto a PVDF membrane with Mini Trans-Blot^®^ Cell using a standard TRIS/glycine 20% methanol buffer, overnight at 35 mV and 4 °C. The free binding site on the PVDF membrane obtained was blocked for 2 h at room temperature under shaking, through tris buffer saline pH 7.6 (TBS) supplied with 5% *w*/*v* non-fat dried milk. After two washing steps with TBS (10 min for each washing), the membrane filters were incubated for 2 h at 37 °C with a solution (1 μg/mL) of primary pAb and mAb antibodies, selected against the recombinant antigen and targeted viruses, diluted in TBS and 1% non-fat dried milk supplied with 0.005% *v*/*v* TWEEN 20. Followed the primary incubation, the membrane was washed three times (10 min each) with TBS and supplied with 0.005% *v*/*v* TWEEN 20 under shaking. Afterwards, the same procedure used for the primary antibodies was performed with secondary antibodies. The PVDF membrane was incubated for 1 h at 37 °C, with a solution (1 μg/mL) of goat anti-rabbit and anti-mouse HRP conjugate diluted in TBS and 5% non-fat dried milk supplied with 0.005% *v*/*v* TWEEN 20. After three washing steps as described above, proteins were visualized by chemiluminescence using the Amersham ECL plus and X-ray films developed manually in the darkroom. A recombinant antigen was used as a positive control.

#### 2.2.3. Indirect and Sandwich ELISA

Indirect ELISA assays were performed to identify appropriate MREs for PIC functionalization. One anti-SIV (Cat. No.MA5-17101, Invitrogen, Waltham, MA, USA) and two anti-ASFV (Cat. No. M.11.PPA.I1BC11 and M.11.PPA.I17AH2, Ingenasa, Madrid, Spain) commercial antibodies were tested. ELISA plates were coated overnight at 4 °C with SIV and ASFV antigens dissolved in 0.1 M bicarbonate/carbonate buffer, pH 9.6. After two washes with PBS containing 0.05% Tween 20, pH 7.4 (PBS-T), plates were blocked using 125 μL/well of PBS + 2.5% Bovine Serum Albumin (BSA) + 0.05% Tween 20, pH 7.4, and incubated for 90 min in room temperature (RT). Plates were washed with PBS-T and after that, 100 μL of anti-SIV and anti-ASF antibodies diluted in PBS + 0.5% BSA + 0.05% Tween 20, pH 7.4, to ratios of 1:500 and 1:1000, were incubated for 90 min in RT. The plates were washed six times with PBST. Secondary HRP-conjugated goat-anti mouse antibody (Cat. No. 62-6520, Invitrogen, Waltham, MA, USA), diluted in a ratio of 1:2000 in PBS + 0.5% BSA + 0.05% Tween 20, pH 7.4, was incubated for 60 min in RT. The plates were washed again six times with PBST and then 100 μL of PBS, pH 7.4, were added for 10 min. Finally, 100 μL of TMB substrate were added and incubated for 10 min at RT. The reaction was finalized using 100 μL of sulfuric acid 0.32 M. Absorbance was measured using a Multiskan™ FC Microplate Photometer (Cat. No. 51119000, Thermo Scientific, Waltham, MA, USA).

For the sandwich ELISA experiments, polyclonal antibodies, used as detection antibodies, were conjugated to the horseradish peroxidase (HRP) enzyme through the Abcam HRP conjugation kit (Cod. ab102890) following the protocol provided by Abcam. The conjugation reaction was performed preparing a solution composed of 100 μL of purified pAb (2 mg/mL), 10μL of modifier reagent provided by the supplier, and 100 μg of HRP lyophilised powder. The pAb/HRP ratio in the reaction was 2:1. The conjugation reaction was conducted in the dark at room temperature (25 °C) for 3 h. Afterwards, 10 μL of quencher reagent (supplied with the kit) was added and the antibodies solution was incubated for 30 min. After this step, the antibodies were ready to use.

The ELISA sandwich assay was performed using the same buffer and timing reported for the indirect ELISA. For the coating step, 50 µL/well of mAb (1 μg/mL) dissolved in 0.1 M bicarbonate/carbonate coating buffer pH 9.6 were deposited. After the washing and blocking step, 50 μL/well of recombinant antigen and/or virus samples diluted in TBS-T, 2% non-fat dried milk were incubated for 2 h at 37 °C. Then, the plate was incubated, for 2 h at 37 °C, with 50 μL/well of HRP-conjugated polyclonal antibodies (1 μg/mL) diluted in TBS-T, 2% non-fat dried milk. After three washing steps, the wells were incubated with TMB substrate (100 μL/well) for 10 min at RT, under shaking. The reaction was stopped with 2.5 M HCl (100 μL/well), and the absorbance was measured using a Tecan Infinite 200 Pro microplate reader (Tecan, Männedorf, Zürich, Switzerland).

Each measurement was performed in triplicate. From the value of the triplicates were calculated the mean and standard deviation.

In the graphs were reported the mean values cleaned from the blank value (no coating in the assay) and values of the error bar were the calculated standard deviation.

### 2.3. Immobilization of the MREs

The protocol for the immobilization of those selected antibodies on the surfaces of the RRs was optimized and the corresponding biosensing tests were carried out. The immobilization protocol included the following steps: (i) surface cleaning/oxidization employing piranha solution (H_2_SO_4_:H_2_O_2_ 3:1 (*v*/*v*)) by means of an immersion bath for 30 min at 100 °C, (ii) surface functionalization employing the organosilane Carboxyethyl Silanetriol disodium salt (25% in water, CTES) at 1% (*v*/*v*) in acidic water (pH 5–6) by means of an immersion bath for 2 h at RT, (iii) chemical activation of CTES layer employing a mixture of 1-Ethyl-3-(3-dimethylaminopropyl)-carbodiimide hydrochloride, and N-Hydroxysuccinimide (EDC/NHS, 2:1) in MES 0.1 M (pH 4) by means of an immersion bath for 30 min at RT, (iv) micro-printing of MREs employing SCENION S1 micro-printing device, and subsequent incubation for 2 h at RT, and (v) PIC surface blocking with fish gelatin 1% in PBS 1x by means of an immersion bath overnight at 4 °C. The biosensor PIC consisted of eight Ring Resonators (RRs), as shown in Figure 1a; four of them were intended for the specific detection of the analytes of interest and were appropriately functionalized for this reason, whereas the other four were used as controls.

Commercial source and catalog number for all employed reagents:−Sulfuric acid ≥98% (H_2_SO_4_, CAS Number 7664-93-9) from VWR (102765G; es.vwr.com/102765G, accessed on 20 September 2021).−Hydrogen peroxide 30% (V/V) (H_2_O_2_, CAS Number 7722-84-1) from Labbox (HYPE-30A-1K0; ien.labbox.com/hydrogen-peroxide, accessed on 20 September 2021).−Carboxyethylsilanetriol disodium salt 25% in water (CTES, CAS Number 18191-40-7) from ABCR (AB110934; abcr.com/AB110934, accessed on 20 September 2021).−1-Ethyl-3-(3-dimethylaminopropyl) carbodiimide hydrochloride (EDC, CAS Number 25952-53-8) from TCI Chemicals (D1601; www.tcichemicals.com/D1601, accessed on 20 September 2021).−N-Hydroxysuccinimide (NHS, CAS Number 6066-82-6) from TCI Chemicals (H0623; www.tcichemicals.com/H0623, accessed on 20 September 2021).−2-(N-morpholino)ethanesulfonic acid (MES, CAS Number 145224-94-8) from Fisher Scientific (BP300-100; www.fishersci.es/10419123/BP300-100, accessed on 20 September 2021).−Gelatine from cold water fish skin (fish gelatine, CAS Number 9000-70-8) from Sigma-Aldrich (G7041, www.sigmaaldrich.com/g7041, accessed on 20 September 2021).−Phosphate Buffered Saline (PBS 10x) from Fisher Scientific (BP399-1; www.fishersci.es/10204733/BP399-1, accessed on 20 September 2021).

### 2.4. Optofluidics for Biosensing

To run the different assays, a microfluidic system with the proper optical access was employed. The silicon chip containing the PIC is housed in a microfluidic system as shown in Figure 2a,b. The chip is attached to the fluidic system by means of a double-sided tape with an open area that, after aligning it with the RRs of the PIC, conforms a fluidic channel on the sensor rings. This design allows the right flowing of the sample of interest just over the sensor components (RRs). The optical signals (input and outputs) are injected to and extracted from the PICs by using a fiber array (with 12 optical fibers) attached to the PIC GCs by using an epoxy resist cured by heat, as shown in Figure 2b. Additionally, this solution satisfies the requirements as a building block of the portable SWINOSTICS device in terms of integration [16].

The SWINOSTICS fluidic module was designed ad hoc to accomplish the desired requirements and manufactured by an external company (microfluidic ChipShop) using an injection molding technique and Cyclo-Oleofin Polymer (COP) as material. The integrated sensing cartridge (PIC + fluidics + optical fibers) was used to perform several tests in the lab to verify the detection capabilities of the sensors. Some preliminary tests were performed by using just a standard straight fluidic channel attached on top of the silicon-based PIC. This was done for simplicity and for saving resources, since the number of manufactured SWINOSTICS microfluidic modules was limited and mainly intended for the field validations. In all cases, the flow rate of the samples was set at 30 µL/min and controlled by a peristaltic pump.

## 3. Results

### 3.1. Identification, Production, and Characterization of the MREs

Commercial pAbs and mAbs, for the selected virus antigens, were identified and purchased, to be employed as MREs for the photonic sensing device. In addition, the recombinant antigen (proteins) was used as a positive control in the immunological and photonic experiments. The selected pAb, mAb, and recombinant antigens are summarized in Table 1.

From each sample (1 mL) of serum or ascites fluid, the obtained yield was ~5 mg and ~2 mg of pure pAb and mAb, respectively. The SDS-PAGE electrophoresis (Appendix A) was used to evaluate antibody purity.

#### 3.1.1. Western Blotting Experiments

The WB experiments were performed to verify the specificity of the selected antibodies. The obtained data reported in Figure 3 show that all of the mAb tested were unable to bind the viruses’ antigens. On the contrary, the selected polyclonal antibodies were able to recognize and bind the antigens.

In detail, the pAb PA5-34969 recognized both the recombinant antigens ORF2 and the virus sample, whereas the mAb M.11.PCV.I36A9 showed a low signal just for the sample virus.

The pAb PRSNP11-S recognized both the NP nucleocapsid protein in the sample virus and the recombinant protein, whereas the mAb SDOW17-A did not recognize neither the NP in the sample line nor the recombinant protein.

As for PPV, two different virus samples (ST and SK-6) were analyzed. The pAb PPVVP21-S recognized the recombinant protein (VP2), and in both sample lines the VP2 capsid protein, whereas the mAb 3C9D11H11 did not work.

The pAb ASFV11-S recognized both the recombinant protein p30 and the viral antigen p30 in the sample virus, whereas the mAb M.11.PPA.I18BB11 did not recognize neither the recombinant protein nor the viral antigen.

About CSFV, both the mAb RAE0826 and the pAb CSFE21-S recognized the recombinant protein E2 but not the E2 capsid viral antigen in the virus sample.

#### 3.1.2. ELISA Test Experiments

ELISA experiments were performed to verify the sensibility (titre) of the selected antibodies. For all target viruses, either indirect or sandwich ELISA assays were performed. As a positive control, inactivated antigens or selected recombinant antigens were used. In addition, the cross-reactivity of pAb selected MREs was evaluated by the ELISA test. The obtained data for both the indirect and sandwich ELISA are presented in Appendix A and summarized in Table 2. From the analysis of the results it is concluded that (i) all of the selected pAb were able to recognize and bind the respective recombinant antigen in a range from 1 up to 0.01 μg/mL except for PRRSV, (ii) the selected pAb were able to recognize PPV, PRRSV, and ASFV, except for PCV2 and CSFV, in the collected sample virus at a different dilution in a range from 1:10 up to 1:1000, and (iii) the mAb anti-ASFV is able to recognize and bind both the recombinant antigen and the viruses in the collected samples, while the rest of the mAb did not work.

Based on these results, the cross-reactivity was evaluated only for the pAb. As reported in Table 2, all possible combinations were explored and tested; in detail, anti-PCV2, anti-PPV, and anti-CSFV results were very specific for the antigen (in fact no cross-reactivity reaction with other antigens was revealed), while the anti-ASFV showed a cross-reactivity against the CSFV antigen and anti-PRRSV showed a cross-reactivity against the PPV and CSFV antigens.

Based on the immunological characterization (ELISA and WB), from the initial identified pAb and mAb the following MREs were selected for the immobilization on the chip surface and the subsequently photonic measurement: pAb-PA5-34969; pAb-PPVVP21-S; pAb-CSFE21-S; pAb-PRSNP11-S; mAb-M.11.PPA.1BC11; mAb-MA5-17101.

### 3.2. Photonic Biosensing

The biosensing experiments required a series of steps to finally obtain the corresponding sensorgrams. The assays for the detection of the targeted swine viruses were performed following a common protocol, which consisted of: (i) flowing the buffer for 5 min, for the acquisition of the photonic signal taken as a baseline; (ii) flowing the sample, which consisted of a suspension of the virus diluted in the same buffer used in the previous step; this was flown for 10 min for the detection of the target; and (iii) rinsing the PIC surface by flowing again the same buffer for 15 min. For the dilution tests, two additional steps were required: (iv) regeneration of the PIC‘s surface by flowing regeneration agent (NaOH and Glycine), for 2 min, and (v) the PIC’s surface was reconditioned by flowing the suitable buffer for 5 min [13].

#### 3.2.1. Targeted Virus PPV

Porcine Parvovirus (PPV—Porcine Parvovirus strain NADL-2 (stock) SK-6 cell cultures, PIWet) sensing experiments, where polyclonal antibodies (pAb-antiPPV, alpha diagnostic PPVVP21-S) at 200 ppm concentration were immobilized on four out of the eight ring resonators, were carried out. The PPV sample was flown over the sensors at dilution factors ranging from 1/100 to 1/5000 in PBS-T buffer. During the whole flowing process, the resonant notch shift of the rings was plotted. Representative results are shown in Figure 4a,b for the lowest and highest dilutions tested, respectively. The plots of the resonant shifts obtained from the RRs used as a reference and those from the RRs with the MREs immobilized showed, as expected, different resonant shifts. The detection signal was obtained as the difference between them.

The detection signals of the sensing experiments for the detection of PPV at the following dilution factors: 1/100, 1/500, 1/1000, 1/2000, and 1/5000 in PBS-T buffer, are shown in Figure 5a. The detection level for each dilution was estimated as the average shift of the data acquired after the rinse step. Results are shown in Figure 5b.

#### 3.2.2. Targeted Virus CSF

Figure 6a shows the plots of the dose dependence of the CSF virus (CSFV—Classical Swine Fever Alfort 106,25 SK6, PIWet) sensing measurement, where pAb-antiCSF (alpha diagnostic CSFE21-S) were immobilized at 200 ppm concentration. The CSF virus was flown over the sensors at dilution factors from 1/100 to 1/5000 in MES buffer. The detection level for each dilution was obtained as the average shift of the data acquired after the rinsing step. Results are shown in Figure 6b. For the highest dilution, quantification of the detection level it is not feasible, and this is marked with an (*).

#### 3.2.3. Targeted Virus PRRSV

The detection signals of the sensing experiments for the detection of PRRSV at the following dilution factors: 1/100, 1/200, 1/500, 1/1000 and 1/5000 in PBS-T buffer, are shown Figure 7a. The detection level for each dilution was estimated as the average shift of the data acquired after the rinse step. Results are shown in Figure 7b.

#### 3.2.4. Targeted Virus ASFV and SIV Virus Sensing Measurements

Aiming at proving that biosensors with monoclonal antibodies against ASFV and SIV microprinted at 100 ppm work properly, virus sensing measurement were carried out for the detection of dilutions of 1/20 and 1/100 in PBS-t + BSA 0.5% for ASFV and SIV, respectively. Figure 8a shows the sensing plots acquired during the assay and Figure 8b shows a bar chart of the shift levels obtained. Both experiments showed the successful detection of the targeted viruses.

## 4. Discussion and Conclusions

Antibodies against the six swine viruses of interest have been successfully characterized and immobilized on the photonic transducers. This achievement along with the development of the proper protocols to carry out on-chip immunoassays allowed the direct detection of six swine viral antigens in buffer at different dilutions, in less than 30 min. Microfluidic channels have been successfully attached to the novel sensors, allowing their exploitation and integration in different setups, such as into the SWINOSTICS device. PICs have been proposed as capable tools for the detection of various analytes such as gases, small molecules and biomolecules [18,19,20]. However, PICs have not yet been proposed as a sensing platform in veterinary diagnostics. To our knowledge, this is the first time that PICs have been exploited to sense pathogens affecting livestock.

In this work, we were able to sense six major viral pathogens at a laboratory setting using a newly developed optofluidic sensor. Each virus provided a different response in terms of shifts in pm. Positive detection was achieved using up to a 1/5000 dilution factor for PCV2 [13] and PPV, both with shifts of about 5 pm, and also for PRRSV, with a shift of 12 pm. For the same dilution, CSFV also showed positive detection potential, but it was not feasible when quantifying it, with a dilution of 1/200 being the lowest quantifiable dilution that gave rise to a shift of 29 pm. Biosensors with monoclonal antibodies against ASFV and SIV also showed positive detection for dilutions of 1/20 and 1/100, with shifts at 9 and 11 pm, respectively. The sensor showed a signal threshold of 5 pm. PICs were proven as a capable platform for the detection of viruses and other analytes.

Future work should focus on further investigating the limit of detection and the limit of quantification, as well as to improve the quantification capabilities of the system. Improved performance of the sensors could be achieved by testing a wider panel of commercial antibodies and immobilizing them in an oriented way. Selection of alternative MREs could pave the way for the detection of different viral antigens and/or circulatory antibodies as well as other analytes.

To conclude, the need for sensors and analytical devices has led to the development of various technological applications targeting a wide range of analytes. However, there are still limited applications for the on-site diagnosis of animal diseases. The novel sensors presented herein prove that PICs can be a powerful platform for the sensitive and specific detection of viral pathogens. Future research can improve the sensing capabilities of the sensors in terms of quantification of the results and the enrichment of the panel of detectable analytes.

## Figures and Tables

**Figure 1 sensors-22-00708-f001:**
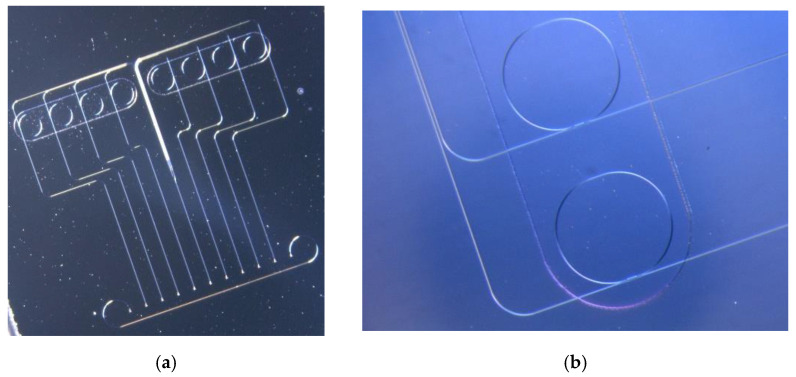
Optical pictures of (**a**) the fabricated biosensor PIC and (**b**) of the PIC close to the region of the RRs, where the window is opened to allow buffer/sample access.

**Figure 2 sensors-22-00708-f002:**
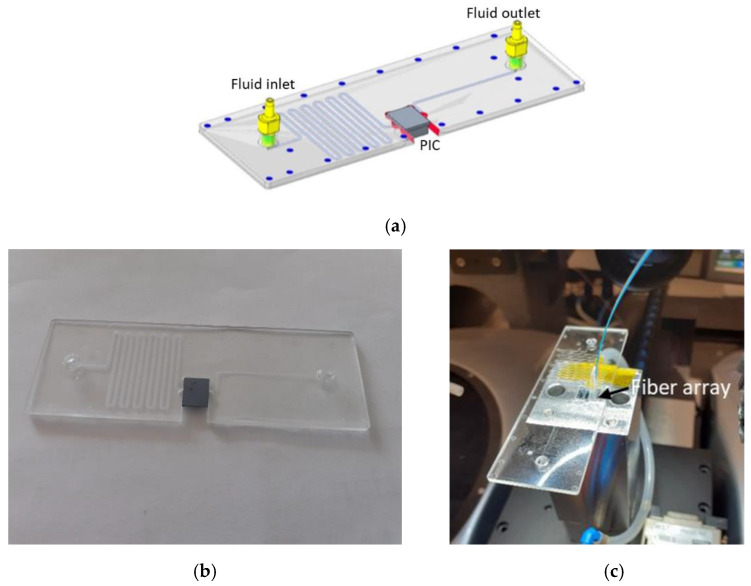
(**a**) 3D sketch of the microfluidic system together with the input and output fluidic ports, and the PIC assembled. (**b**) Bottom view of the silicon-based PIC assembled into the microfluidic system. (**c**) Representative picture of the optofluidic system; for the sake of clarity the required tubing to connect the microfluidic channel with the pump and the analyte container to handle the fluidics was not connected.

**Figure 3 sensors-22-00708-f003:**
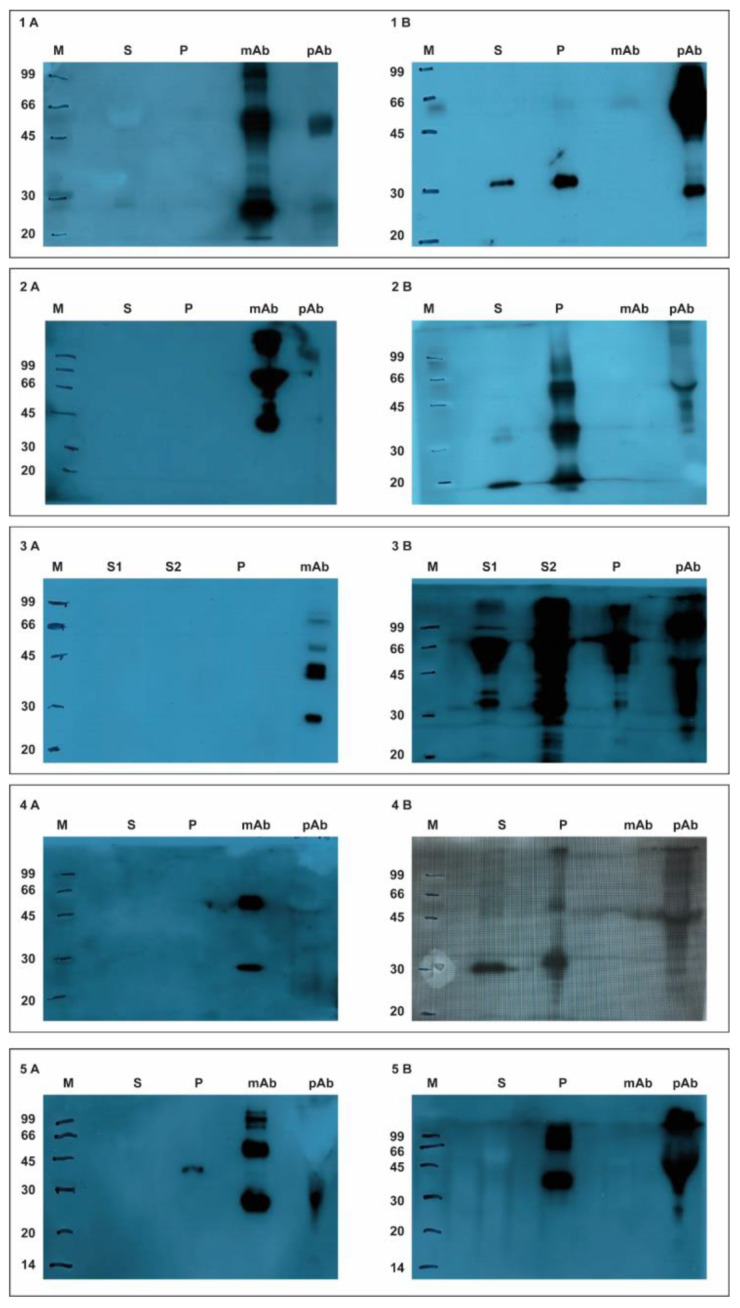
Panel (**1A**): WB results of the mAb anti-PCV2 against virus sample (line S), recombinant ORF2 capsid protein (line P), and purified mAb and pAb anti-PCV2 (lines mAb and pAb). Panel (**1B**): WB results of the pAb anti-PCV2 against virus sample (line S), recombinant ORF2 capsid protein (line P), and purified mAb and pAb anti-PCV2 (lines mAb and pAb). Panel (**2A**): WB results of the mAb anti-PRRSV against virus sample (line S), recombinant NP nucleocapsid protein (line P), and purified mAb and pAb anti-PRRSV (lines mAb and pAb). Panel (**2B**): WB results of the pAb anti-PRRSV against virus sample (line S), recombinant NP nucleocapsid protein (line P), and purified mAb and pAb anti-PRRSV (lines mAb and pAb). Panel (**3A**): WB results of the mAb anti-PPV against ST and SK-6 virus samples (lines S1 an S2), recombinant _VP2 protein (line P), and purified mAb and pAb anti-PPV (lines mAb and pAb). Panel (**3B**): WB results of the pAb anti-PPV against ST and SK-6 virus samples (lines S1 an S2), recombinant VP2 protein (line P), and purified mAb and pAb anti-PPV (lines mAb and pAb). Panel (**4A**): WB results of the mAb anti-ASFV against virus sample (line S), recombinant p30 capsid protein (line P), and purified mAb and pAb anti-ASFV (lines mAb and pAb). Panel (**4B**): WB results of the pAb anti-ASFV against virus sample (line S), recombinant p30 capsid protein (line P), and purified mAb and pAb anti-ASFV (lines mAb and pAb). Panel (**5A**): WB results of the mAb anti-CSFV against virus sample (line S), recombinant E2 envelope protein (line P), and purified mAb and pAb anti-CSFV (lines mAb and pAb). Panel (**5B**): WB results of the pAb anti-CSFV against virus sample (line S), recombinant E2 envelope protein (line P), and purified mAb and pAb anti-CSFV (lines mAb and pAb).

**Figure 4 sensors-22-00708-f004:**
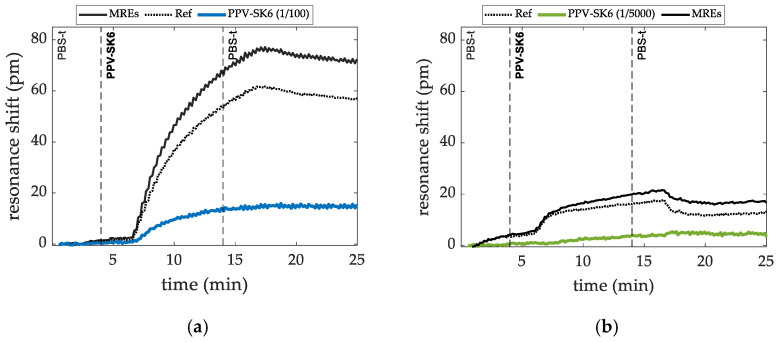
Differences between resonant notch shift obtained from the RRs with MREs immobilized on top of the sensors (black line) and from the references (dotted line) for the detection of PPV at dilutions of PPV at 1/100 (**a**) and at 1/5000 (**b**).

**Figure 5 sensors-22-00708-f005:**
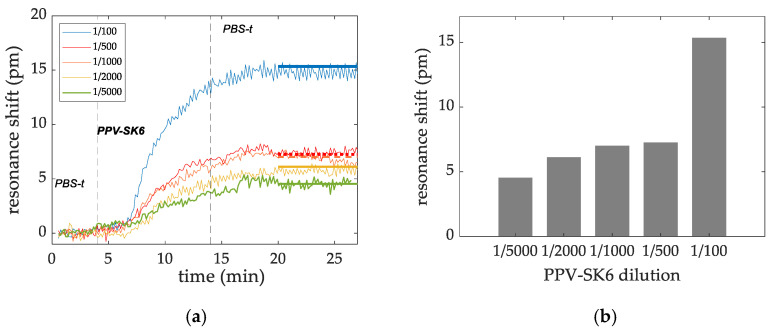
(**a**) Dose dependence signal shift (measured in pm) for pAb (alpha diagnostic) for each PPV dilution tested. The detection level for each dilution was obtained as the average shift of the data acquired after the rinse step and depicted as a horizontal line. (**b**) The detection levels for each dilution tested are summarized in a bar chart.

**Figure 6 sensors-22-00708-f006:**
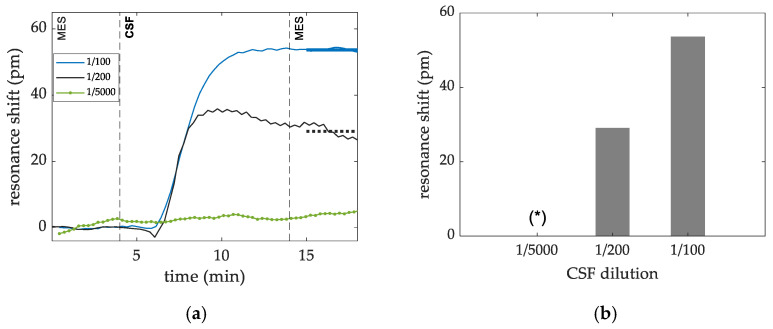
(**a**) Dose dependence signal shift (measured in pm) for pAb (alpha diagnostic) for each CSFV dilution tested. The detection level for each dilution was obtained as the average shift of the data acquired after the rinsing step and depicted as a horizontal line. (**b**) The detection levels for each dilution tested are summarized in a bar chart. For the highest dilution, quantification of the detection level it is not feasible, and this is marked with an (*).

**Figure 7 sensors-22-00708-f007:**
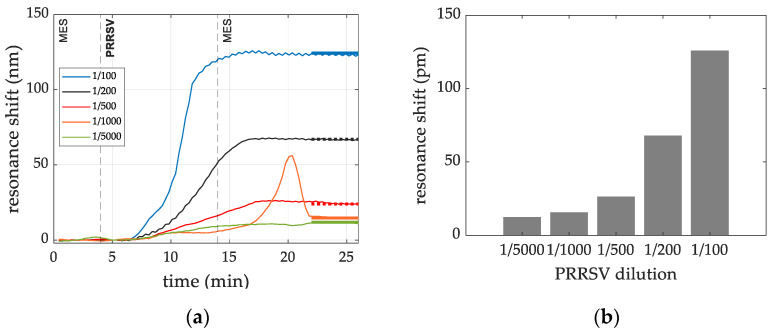
(**a**) Dose dependence signal shift (measured in pm) for pAb (alpha diagnostic) for each PRRSV dilution factor. The peak observed at minute 20 for the (1/1000) dilution involves a RI increase. It is ascribed to fluidic instabilities in the fluidic chamber, where the two solutions are being exchanged, giving rise to an increase of the refractive index [17]. (**b**) The detection levels for each dilution tested are summarized in a bar chart.

**Figure 8 sensors-22-00708-f008:**
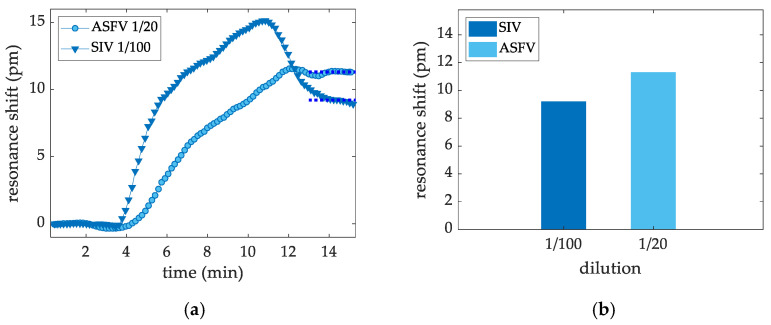
(**a**) Signal shift (measured in pm) obtained for the detection of ASFV and SIV viruses at 1/20 and 1/100 dilutions in PBS-T + BSA 0.5%, respectively. (**b**) The detection levels for each virus tested are summarized in a bar chart.

**Table 1 sensors-22-00708-t001:** Summary of the viruses targeted, the recombinant antigens, and the corresponding MREs for their proper detection. (*) Dilutions tests have already been reported [13].

Target	Recombinant Antigen	MREs
Porcine Circovirus—PCV2(*)	Capsid protein ORF2 (30 kDa) alpha diagnostic PCV2C25-R-10	pAb—Thermo Fisher PA5-34969 mAb—Ingenasa M.11.PCV.I36A9
Porcine Reproductive and Respiratory Syndrome—PRRSV	Nuclear protein NP (15 kDa) alpha diagnostic PRSNP-15-R-10	pAb—alpha diagnostic PRSNP11-S mAb—RTI, LLC SDOW17-A
Porcine Parvovirus—PPV	VP2 protein (68 kDa) alpha diagnostic PPVVP21-R-10	pAb—alpha diagnostic PPVVP21-S mAb—VMRD 3C9D11H11
African Swine Fever Virus—ASFV	p30 protein 24.4 kDa alpha diagnostic ASFV15-R-10	pAb—alpha diagnostic ASFV11-S mAb—Ingenasa M.11.PPA.I18BB11 mAb—Ingenasa M.11.PPA.1BC11
Classical Swine Fever Virus—CSFV	Envelope protein E2 (35 kDa) alpha diagnostic CSFE25-R-10	pAb—alpha diagnostic CSFE21-S mAb—alpha scientific RAE0826
Swine Influenza A Virus—SIV		mAb—Thermo Fisher MA5-17101

**Table 2 sensors-22-00708-t002:** Summary of results obtained by indirect and sandwich ELISA tests.

Virus		Indirect ELISA		Sandwich ELISA	Cross-Reactivity of Polyclonal Antibodies				
		*pAb*	*mAb*	*mAb*/*pAb*	*Anti-PCV2*	*Anti-* *PPV*	*Anti-PRRSV*	*Anti-CSFV*	*Anti-ASFV*
PCV2	*S*	(−)	(−)	(−)	(−)	(−)	(−)	(−)	(−)
	*P*	√	(−)	(−)	√	(−)	(−)	(−)	(−)
PRRSV	*S*	√	(−)	(−)	(−)	(−)	√	(−)	(−)
	*P*	(−)	(−)	(−)	(−)	(−)	(−)	(−)	(−)
PPV	*S*	√	(−)	(−)	(−)	√	√	(−)	(−)
	*P*	√	(−)	(−)	(−)	√	√	(−)	(−)
ASFV	*S*	√	√	(−)	(−)	(−)	(−)	(−)	√
	*P*	√	√	(−)	(−)	(−)	(−)	(−)	√
CSFV	*S*	(−)	(−)	(−)	(−)	(−)	√	(−)	(−)
	*P*	√	(−)	(−)	(−)	(−)	(−)	√	√
SIV		(−)	√	(−)					

(−) = negative result. √ = positive result.

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
