# Peer review of "Photonic Label-Free Biosensors for Fast and Multiplex Detection of Swine Viral Diseases"

_sensors, 2022, doi:10.3390/s22030708_

Round 1

Reviewer 1 Report

The presented paper offers an interesting insight on photonic biosensors based on resonant microrings for swine viral diseases.

The text is well written, the presented topic is well introduced. the investigation method is very well described and the results are many, well presented and completely understandable. This work is well-suited for publication in Sensors.

My first suggestion is to improve the quality of Figure 1 and Figure 2: the presented real-world pictures are perflectly fine, but if also 2D or 3D sketches could be included in the figures, the reader would appreciate and understand more the design (especially in Figure 2).

Another minor suggestion is to rearrange the periods in the "Discussion and conclusions" final section. As an example, I feel like the period going from line 404 to line 412 should belong more to an introductory section.

Thank you for your patience and Merry Christmas to all the authors

Reviewer 2 Report

Please find comments below-

Section 2.3 Immobilization of the MREs-Details on the protocol such as duration for which surface cleaning, surface functionalization/activation is performed, volume used & methodology (was liquid pipetted on etc.). Since these will affect reproducibility of this step. Please also provide catalog number & source for all reagents such as NHS/EDC/MES buffer & others are appropriate (some of them have been already described).

Line 219- Incorrect sentence- "When attaching the chip to the fluidic system forms a fluidic channel which allows to expose the sensors to the sample of interest"

Line 220- Please clarify this sentence better. "For the input and output of optical 220 signals trough a fiber array, in the front part of the microfluidic system there is an open 221 window coincident with the GCs, as shown in Figure 2(b)."

Line 224-"The designed SWINOSTICS fluidic module was fabricated by an external company (ChipShop microfluidic)."- If this is a ChipShop catalog item please provide reference. If not please describe method of fabrication/material with microfluidic design/drawings (maybe in supporting info). Or if this is similar to previous paper please provide ref. for the readers.

Please label Figure 2(b) or it's schematic equivalent (with say fludic inlet/outlet, fiber array)

Line 227-"sensors. Some of the tests were performed by using a small and simple fluidic system, just a standard straight fluidic channel attached on top of the silicon-based 228 PIC."

Round 2

Reviewer 1 Report

Accept it in its current form